# FN9-10ELP, an ECM-Mimetic Fusion Protein, Protects Human Mesenchymal Stem Cells from Etoposide-Induced Senescence

**DOI:** 10.3390/ijms26189218

**Published:** 2025-09-21

**Authors:** Su-Hyeon Jang, Jun-Hyeog Jang

**Affiliations:** Department of Biochemistry, School of Medicine, Inha University, Incheon 22212, Republic of Korea; suhJang@inha.edu

**Keywords:** cellular senescence, hMSCs, FN9-10ELP, ECM-mimetic biomaterial, etoposide, SASP, fibronectin, elastin-like polypeptide

## Abstract

Cellular senescence is a major barrier to the therapeutic application of human mesenchymal stem cells (hMSCs), as it compromises their proliferative capacity, differentiation potential, and regenerative efficacy. In this study, we investigated whether FN9-10ELP, a recombinant extracellular matrix (ECM)-mimetic fusion protein composed of fibronectin type III domains 9 and 10 conjugated to elastin-like polypeptides (ELPs), could attenuate etoposide-induced senescence in human turbinate-derived MSCs (hTMSCs). Premature senescence was induced by treatment with 20 µM etoposide, and the protective effects of FN9-10ELP were evaluated in terms of cell viability (using the MTT assay), senescence-associated gene expression (by RT-qPCR analysis), nuclear morphology (after staining with 4’,6-diamidino-2-phenylindole (DAPI)), and SA-β-galactosidase activity. FN9-10ELP treatment significantly improved cell viability and reduced the expression of *senescence-associated secretory phenotype (SASP)* genes, including *interleukin-6 (IL-6)*, *interleukin-8 (IL-8),* and *plasminogen activator inhibitor-1 (PAI-1).* Furthermore, FN9-10ELP alleviated nuclear enlargement and decreased the proportion of SA-β-gal-positive cells, indicating suppression of the senescence phenotype. These findings demonstrate that FN9-10ELP effectively counteracts chemotherapy-induced senescence in hMSCs and highlight its potential as a promising biomaterial for regenerative medicine and anti-aging therapies.

## 1. Introduction

Human mesenchymal stem cells (hMSCs) are widely recognized as a promising cell source for regenerative medicine owing to their multipotent differentiation potential, immunomodulatory properties, and tissue repair ability [1]. However, their clinical application is restricted by cellular senescence, which can occur during long-term in vitro expansion or under genotoxic and oxidative stress conditions [2,3]. Senescent hMSCs exhibit impaired proliferative capacity, reduced differentiation potential, and diminished immunomodulatory function, ultimately leading to decreased therapeutic efficacy [2,3]. Importantly, senescent hMSCs are not only induced by artificial culture conditions but are also observed in aged or chronically inflamed tissues in vivo, where they contribute to tissue dysfunction and age-related diseases [4]. Hallmarks of senescent hMSCs include increased cell size, elevated SA-β-galactosidase (SA-β-gal) activity, and enhanced secretion of pro-inflammatory cytokines and matrix-degrading enzymes, collectively referred to as the senescence-associated secretory phenotype (SASP) [4,5]. These findings highlight the urgent need to develop strategies to inhibit or delay hMSCs senescence.

Extracellular matrix (ECM)-mimetic biomaterials have recently gained attention due to their ability to recapitulate the native microenvironment and regulate stem cell behavior [5]. Beyond serving as a structural scaffold, the ECM delivers biochemical and mechanical cues via integrin-mediated signaling, thereby orchestrating essential cellular processes including DNA repair, oxidative stress responses, and inflammation [6,7]. Accumulating evidence indicates that alterations in ECM composition or stiffness can activate pathways such as focal adhesion kinase (FAK), mitogen-activated protein kinase (MAPK), and phosphatidylinositol 4,5-bisphosphate 3-kinase/AKT (PI3K/AKT), thereby promoting cellular senescence [8,9]. Conversely, restoring ECM-like conditions has been shown to reduce senescence markers and promote cellular rejuvenation [10].

In this study, the key material used, FN9-10ELP, is an ECM-mimetic fusion protein composed of the type III 9th and 10th repeats of human fibronectin (FN9-10, containing the integrin-binding motifs PHSRN and RGD) fused with an elastin-like polypeptide (ELP) consisting of the repeating pentapeptide sequence VPGXG (where X is any amino acid except proline) [11,12,13,14,15]. The FN9-10 domain mediates binding to integrin α5β1 through the PHSRN motif in FN9 and the RGD motif in FN10, thereby activating signaling pathways that regulate cell adhesion, proliferation, and survival, whereas the ELP possesses high biocompatibility, biodegradability, and low immunogenicity, making it a promising platform for diverse biomedical applications including drug delivery, tissue engineering, and surface modification [14,15,16]. The FN9-10 domains were derived from human fibronectin, and the ELP sequence was de novo designed based on consensus elastin motifs, was expressed in *Escherichia coli* via recombinant DNA technology, and was purified using inverse transition cycling [17]. This modular design endows FN9-10ELP with excellent biocompatibility, structural stability, thermoresponsive behavior, and bioactivity, enabling it to mimic native ECM functions and modulate cellular adhesion, proliferation, and survival. Previous studies have demonstrated that FN9-10ELP enhances hMSCs adhesion and proliferation, contributing to the maintenance of stem cell functionality [17,18]. However, whether FN9-10ELP can mitigate hMSCs senescence under DNA dam-age-induced stress conditions remains unexplored.

For this purpose, we employed human turbinate-derived MSCs (hTMSCs), which are an attractive stem cell population due to their ease of isolation, robust proliferative capacity, and multipotent differentiation potential into osteogenic, chondrogenic, and adipogenic lineages [19,20]. In addition, hTMSCs exhibit unique immunomodulatory functions by regulating cytokine secretion in response to Toll-like receptor (TLR) activation, thereby broadening their therapeutic potential [21]. Nevertheless, like other hMSCs populations, hTMSCs remain vulnerable to premature senescence induced by external stressors, including DNA damage [22].

Premature senescence triggered by DNA damage represents a well-established cellular stress response [23], and the etoposide-induced senescence model is widely used to experimentally recapitulate this process [24,25]. Etoposide, a DNA topoisomerase II inhibitor, is broadly applied as both an anticancer drug and an experimental tool for senescence induction. By generating persistent DNA double-strand breaks (DSBs) and activating the DNA damage response (DDR), etoposide reliably induces a premature senescence phenotype [26,27]. Thus, etoposide treatment provides a robust and clinically relevant model for investigating anti-senescence strategies in hMSCs.

Therefore, the present study aimed to investigate whether FN9-10ELP, an ECM-based biomaterial, could attenuate etoposide-induced cellular senescence in hMSCs. To evaluate its protective effects, we assessed cell viability, expression of senescence-associated genes, nuclear morphology, and SA-β-gal activity. We hypothesized that FN9-10ELP would alleviate and protect against cellular senescence by recreating ECM-like conditions and promoting integrin-mediated signaling. Our findings suggest that FN9-10ELP can preserve the functional properties of hMSCs and mitigate senescence, underscoring its potential as a promising biomaterial for regenerative medicine and anti-aging therapies.

## 2. Results

### 2.1. FN9-10ELP Protects hMSCs from Etoposide-Induced Cellular Senescence by Promoting Proliferation

To evaluate whether the recombinant ECM-mimetic fusion protein FN9-10ELP can attenuate etoposide-induced senescence in hMSCs, cell proliferation was assessed using the MTT assay (Figure 1). On day 0, cells were seeded onto FN9-10ELP-coated or uncoated control group, followed by etoposide treatment, and the MTT assay was performed 1 h post-treatment. At this early time point, there was no significant difference in absorbance between the FN9-10ELP group and the control group, indicating comparable initial seeding conditions rather than established cell adhesion. However, on day 1, the absorbance of the FN9-10ELP group was significantly higher than that of the control group (*p* < 0.05), and this effect became even more pronounced on day 2 (*p* < 0.001). These results demonstrate that FN9-10ELP effectively enhances the proliferation of hMSCs in a time-dependent manner under etoposide-induced senescence conditions and indirectly suggest its role in attenuating cellular senescence. Notably, the anti-senescent effect of FN9-10ELP was evident as early as day 1 and was further enhanced by day 2.

### 2.2. FN9-10ELP Suppresses the Expression of Senescence-Associated Genes in Etoposide-Treated hMSCs

To evaluate the anti-senescent effect of FN9-10ELP, hMSCs were cultured on 100 mm dishes coated with FN9-10ELP and non-coated (control), followed by treatment with 20 μM etoposide for two days to induce cellular senescence. The mRNA expression levels of the senescence-associated genes *IL-6, IL-8,* and *PAI-1* were measured using real-time PCR, with *β-actin* used as an internal control. As a result, the expression of all three genes was significantly decreased in the FN9-10ELP condition. Both *IL-6* and *IL-8* were significantly downregulated following FN9-10ELP treatment (*p* < 0.05), and *PAI-1* expression showed a more significant reduction (*p* < 0.01) (Figure 2). These results suggest that FN9-10ELP effectively suppresses the expression of senescence-associated genes under etoposide-induced stress conditions, supporting its potential role in modulating cellular senescence in hMSCs.

### 2.3. FN9-10ELP Attenuates Nuclear Enlargement Induced by Etoposide

To evaluate whether FN9-10ELP can attenuate etoposide-induced cellular senescence, we analyzed nuclear morphology using DAPI staining (Sigma, St. Louis, MO, USA). This method enables visualization of senescence-associated changes such as nuclear enlargement, irregular shape, and altered chromatin organization. Representative images (Figure 3A) depict the overall nuclear morphology of hMSCs in the control and FN9-10ELP-coated groups after etoposide exposure. Although qualitative differences appeared subtle at this magnification, quantitative morphometric analysis (Figure 3B) revealed a significant reduction in the mean nuclear area in the FN9-10ELP-coated group compared with the control group. Specifically, the mean nuclear area increased to 4338.45 μm^2^ in the control group, but was significantly smaller at 3438.61 μm^2^ in the FN9-10ELP group (*p* < 0.001). Nuclear area was quantified from at least 100 randomly selected nuclei per condition across three independent experiments to ensure statistical robustness. Collectively, these findings demonstrate that FN9-10ELP alleviates nuclear enlargement, a hallmark of cellular senescence, and underscore its potential as a functional biomaterial to mitigate etoposide-induced senescence in hMSCs.

### 2.4. FN9-10ELP Significantly Suppresses Etoposide-Induced SA-β-Galactosidase Activity in Senescent hMSCs

To further evaluate the anti-senescence effect of FN9-10ELP, we performed SA-β-gal staining, a widely used biomarker of cellular senescence reflecting increased lysosomal β-galactosidase activity at pH 6.0. hMSCs were treated with 20 μM etoposide for 5 days under control or FN9-10ELP-coated conditions, followed by staining according to the manufacturer’s instructions. Images were acquired under identical microscope settings with a fixed scale bar of 100 μm to allow direct comparison. Quantification was performed using Fiji software (ImageJ, version 2.3.0, NIH, Bethesda, MD, USA), with positive cells defined as those showing distinct blue cytoplasmic staining above background. At least 200 cells per condition were analyzed, pooled from three independent experiments. Etoposide markedly increased the percentage of SA-β-gal-positivity cells in the control group, whereas FN9-10ELP treatment significantly reduced this proportion (*p* < 0.01). Collectively, these results demonstrate that FN9-10ELP effectively mitigates etoposide-induced cellular senescence in hMSCs.

## 3. Discussion

Cellular senescence has been extensively investigated in the broader contexts of organismal aging, tissue dysfunction, and stem cell exhaustion [2,3,4]. However, the impact of senescence on mesenchymal stem cells (MSCs)—cells that play critical roles in tissue regeneration, immunomodulation, and repair—has only in recent years received increasing attention in the context of its clinical relevance [5,6,7]. Senescence in MSCs can be triggered by a variety of stressors, including telomere attrition, oxidative stress, DNA damage, and chronic inflammatory signaling [8,9,10], leading to permanent cell cycle arrest and the acquisition of a senescence-associated secretory phenotype (SASP) characterized by elevated secretion of pro-inflammatory cytokines, chemokines, and matrix-degrading enzymes [11,12,13]. This senescence-associated reprogramming not only impairs the regenerative and differentiation potential of hMSCs [14,15] but also contributes to a pro-inflammatory microenvironment that exacerbates tissue degeneration in vivo [1,16]. Notably, hMSCs senescence has been implicated in reduced therapeutic efficacy of cell-based interventions for degenerative and inflammatory disorders [19,20], underscoring the clinical importance of developing strategies to delay or reverse senescence in these cells. In this study, we investigated the effects of FN9-10ELP, a recombinant ECM-mimetic fusion protein, on etoposide-induced senescence in MSCs, providing mechanistic insights into how ECM-based interventions may mitigate senescence-associated decline in stem cell function.

The observed increase in cell viability in FN9-10ELP-treated hMSCs under etoposide-induced senescence suggests that FN9-10ELP confers a protective effect against DNA damage-induced cellular dysfunction. This effect may be attributed to FN9-10’s integrin-binding domains (RGD and PHSRN motifs), which can activate pro-survival signaling cascades such as FAK/PI3K/AKT and MAPK pathways, thereby enhancing cellular resistance to stress-induced senescence. Previous studies have reported that FN9-10ELP promotes hMSCs adhesion and proliferation on both polymeric and metallic surfaces [17,18], supporting the idea that enhanced adhesion-mediated signaling may contribute to the maintenance of metabolic activity and the delay of senescence.

To assess whether FN9-10ELP, a recombinant ECM-mimetic fusion protein, could suppress cellular senescence in hMSCs induced by etoposide (20 μM), cell viability was evaluated using the MTT assay (Figure 1). On day 0, no significant difference in absorbance was observed between the FN9-10ELP-coated group and the non-coated group, indicating similar levels of initial cell attachment and viability. However, by day 1, the FN9-10ELP group showed significantly higher absorbance values compared to the control (*p* < 0.05), and this difference became even more pronounced on day 2 (*p* < 0.001). These findings suggest that FN9-10ELP enhances the viability of hMSCs under DNA damage-induced senescent conditions in a time-dependent manner, possibly by preserving cellular metabolic activity and increasing resistance to apoptosis or senescence. Notably, the significant difference observed on day 2 indicates a cumulative anti-senescent effect of FN9-10ELP over time. Lee et al. (2019) reported that FN9-10ELP-coated plates promote adhesion and proliferation of MSCs [17], and Park et al. (2021) demonstrated that FN9-10ELP coating on titanium surfaces significantly enhanced hMSCs attachment and proliferation [18]. These reports are consistent with our findings, further supporting the notion that FN9-10ELP protects hMSCs from etoposide-induced stress and promotes cell survival. Given that hMSCs are pivotal in tissue regeneration and anti-aging research, the observed enhancement of their viability by FN9-10ELP underscores its therapeutic potential.

RT-qPCR analysis revealed that the expression of senescence-associated genes—*IL-6*, *IL-8*, and *PAI-1*—was significantly reduced in FN9-10ELP-coated hMSCs compared to non-coated controls (Figure 2). *PAI-1* is a key factor known to amplify cellular senescence by enhancing the production of SASP components such as *IL-6* and *IL-8* [28], while *IL-6* and *IL-8* are representative SASP markers that are induced by DNA damage and contribute to proinflammatory microenvironments [29,30]. Chemotherapeutic agents, such as etoposide, are known to activate NF-κB, which in turn increases the levels of *IL-6* and *IL-8* [31,32]. Additionally, NF-κB can act as a negative regulator of PTEN, modulating downstream AKT signaling [33]. FN9-10ELP may suppress etoposide-induced NF-κB activation and SASP expression, thereby contributing to the reduction in senescence markers observed in hMSCs. Thus, this downregulation suggests that FN9-10ELP can alleviate cellular senescence by inhibiting inflammatory signaling pathways.

Quantitative analysis of nuclear size via DAPI staining further supported the anti-senescent effects of FN9-10ELP. hMSCs cultured on FN9-10ELP-coated surfaces displayed significantly smaller nuclear areas than those on non-coated surfaces (Figure 3). Nuclear and cellular enlargement is a well-established morphological hallmark of senescent cells. Indeed, Pathak et al. (2021) reported abnormal nuclear morphology and enlargement in senescent cells [34], and Heckenbach et al. (2022) quantitatively demonstrated increased nuclear size in both replicative and ionizing radiation-induced senescence models [35]. Therefore, the reduced nuclear area observed in FN9-10ELP-coated conditions suggests a delay or attenuation of senescence-associated morphological alterations.

Consistent with this, SA-β-gal staining revealed that the proportion of SA-β-gal-positive cells was significantly lower in the FN9-10ELP-coated group compared to the control (Figure 4). SA-β-galactosidase is a well-established biomarker of cellular senescence [36], and the reduced SA-β-gal activity observed here supports the anti-senescent effect of FN9-10ELP. Similarly, Tragoonlugkana et al. (2024) reported that adipose-derived stem cells (ADSCs) cultured on fibronectin (FN) or vitronectin (VN)-coated surfaces exhibited significantly fewer SA-β-gal-positive cells compared to non-coated controls [37]. Although different cell types were used, these findings are in agreement with our results and support the notion that ECM protein coatings can attenuate stem cells senescence. Hence, FN9-10ELP, an ECM-mimetic protein, demonstrates potential as a functional biomaterial for senescence attenuation. In addition, the PI3K/AKT signaling pathway is known to regulate lysosomal biogenesis and autophagy, key processes underlying SA-β-gal activity [38]. AKT activation regulates mTORC1 activity, which in turn prevents excessive lysosomal enlargement and maintains autophagic flux [39]. As mTORC1 functions as a central regulator of lysosomal capacity and autophagy, the activation of AKT may contribute to the normalization of lysosomal homeostasis and the preservation of autophagic balance in senescent cells [39]. Therefore, the observed reduction in SA-β-gal-positive cells following FN9-10ELP treatment may be partially attributable to integrin-mediated activation of AKT, which could potentially contribute to the restoration of lysosomal homeostasis and the mitigation of senescence-associated cellular alterations. These findings indirectly suggest a mechanistic connection between ECM-mediated signaling, AKT activation, and SA-β-gal activity in senescent cells.

Although we did not directly measure intracellular ROS levels in this study, previous research has shown that etoposide induces DNA damage, mitochondrial biogenesis, and cytotoxicity through excessive ROS generation [40]. Moreover, ECM proteins containing fibronectin domains similar to FN9-10 have been reported to modulate oxidative stress and reduce ROS accumulation in MSCs under stress conditions [41]. Taken together, these findings suggest that FN9-10ELP may help maintain redox homeostasis and protect hMSCs from oxidative stress-induced senescence, consistent with the anti-senescent effects observed in our study.

Our findings suggest that FN9-10ELP not only serves as a mechanical support for cell attachment, but also modulates intracellular signaling to suppress cellular senescence. The FNIII9-10 domain is known to interact with integrin α5β1 or αvβ3 through its RGD motif, thereby activating signaling pathways such as FAK, PI3K/AKT, and MAPK/p38, which are involved in promoting cell survival, regulating inflammation, and delaying senescence [42,43]. FN9-10ELP likely exerts its effects through modulation of these pathways, contributing to SASP attenuation and suppression of senescence-related morphological changes.

Moreover, the ELP domain enhances protein stability and surface adsorption, thereby improving the biological activity of the FN9-10 domain. For instance, FN9-10ELP fusion proteins have been shown to promote hMSCs adhesion and proliferation [17], and ELP derivatives containing RGD or YIGSR ligands have been reported to enhance fibroblast adhesion and growth [16]. These findings suggest that ELP improves cell-ECM interactions, thereby enhancing stem cell function and delaying senescence.

In terms of quality and reliability, multiple complementary assays—MTT viability, SA-β-gal staining, nuclear morphology via DAPI staining, and senescence-associated gene expression—were used to evaluate senescence, allowing cross-validation of findings across independent experimental endpoints. All experiments were performed in triplicate using hMSCs from multiple donors, and quantitative image analyses were carried out using standardized software (Fiji/ImageJ, NIH) with fixed thresholds to minimize observer bias. Statistical significance was determined using Welch’s *t*-test, and data were presented as mean ± SD to accurately reflect variability.

However, this study has limitations. To assess the preventive effect of FN9-10ELP on etoposide-induced senescence, hMSCs were first seeded onto FN9-10ELP-coated or un-coated control dishes, followed by treatment with 20 μM etoposide. This approach al-lowed us to specifically evaluate the preventive action of FN9-10ELP against senescence induction. In this study, all experimental groups were subjected to etoposide treatment to induce senescence, and the effects of FN9-10ELP were evaluated under these conditions. This design enabled us to specifically assess the preventive effect of FN9-10ELP against DNA damage-induced senescence. Untreated young hMSCs were not included as additional controls, since their baseline characteristics and responses under normal culture conditions have already been established in our previous studies [17]. Nonetheless, the absence of young counterparts in the present work is acknowledged as a limitation, and future studies should incorporate untreated and FN9-10ELP-treated young hMSCs to more comprehensively distinguish anti-senescence effects from general anti-inflammatory activity. We acknowledge, however, that this experimental design does not allow us to assess whether FN9-10ELP can reverse an already established senescent phenotype. Future studies using pre-senescent hMSCs, in which cells are first driven to senescence and then cultured in the presence of FN9-10ELP, would be valuable to determine whether FN9-10ELP can ameliorate established senescence. Furthermore, additional research is needed to clarify the molecular mechanisms of FN9-10ELP, particularly its effects on integrin and downstream signaling pathways. In addition, the long-term impact of FN9-10ELP on stem cells stemness and differentiation potential remains to be investigated.

Nevertheless, this study demonstrates that FN9-10ELP can effectively suppress hMSCs senescence and promote cell proliferation. Previous studies have reported that FN9-10ELP enhance hMSCs attachment and proliferation when coated on titanium surfaces [18], supporting its potential applicability in tissue engineering. While these findings are encouraging, confirming their full translational relevance requires addressing remaining methodological limitations and performing additional validation in broader experimental settings. Future studies should evaluate the applicability of FN9-10ELP on diverse biomaterial surfaces (e.g., hydrogels, polymer scaffolds) and assess its performance in in vivo models. Taken together, the present data suggest that FN9-10ELP, as an ECM-mimetic biomaterial, holds potential in stem cell-based tissue regeneration and anti-aging therapeutics.

## 4. Materials and Methods

### 4.1. Construction and Expression of Recombinant FN9-10ELP Fusion Protein

As previously described [17], an ELP coding sequence was designed using Val, Leu, and Gly at a 17:4:9 ratio as guest residues. The full ELP[V17L4G9] sequence, synthesized by Genotech (Daejeon, Republic of Korea), incorporated these residues at the fourth position (Xaa) of the pentapeptide repeat (Val-Pro-Gly-Xaa-Gly). The sequence was amplified by PCR and cloned into the pBAD-His-FNIII9-10 vector using *Sac*I and *Hind*III restriction sites. The resulting construct was transformed into *E. coli* TOP10 for recombinant protein expression. For expression, transformed *E. coli* were cultured overnight at 37 °C in Luria Broth containing ampicillin (LB-Amp). When cultures reached an OD_600_ of 0.6, protein expression was induced by the addition of 0.1% (*w*/*v*) L-arabinose. After 6 h of induction, cells were harvested by centrifugation at 6000× *g* for 10 min, lysed, and sonicated. The soluble fraction was collected by centrifugation at 13,000× *g* for 25 min at 4 °C. Crude FN9-10ELP fusion protein was isolated by adding 3 M NaCl to the supernatant, incubating at 40 °C, and centrifuging at the same temperature. The resulting pellet was resuspended in ice-cold 1× phosphate-buffered saline (PBS), followed by a final centrifugation at 4 °C to increase purity. Protein purity was confirmed by 12% SDS-PAGE and Coomassie Brilliant Blue staining.

### 4.2. hMSC Culture and Preparation

hMSCs were derived from nasal inferior turbinate tissues obtained from patients prior to surgery. This study was approved by the Institutional Review Board (IRB) of the Catholic University Seoul St. Mary’s Hospital (approval number KC08TISS0341). Detailed methods for isolation and culture of hMSCs were have been previously reported. Briefly, the previous work confirmed the mesenchymal stem cell identity of hMSCs through flow cytometric analysis, demonstrating high expression of MSC markers (CD73, CD90, CD105) and absence of hematopoietic markers (CD34, CD45, HLA-DR), as well as through successful osteogenic, chondrogenic, and adipogenic differentiation [44].

hMSCs were cultured in α-minimal essential medium (α-MEM; Welgene, Gyeongsan, Republic of Korea) supplemented with 10% fetal bovine serum (FBS; Welgene, Gyeongsan, Republic of Korea), 100 μg/mL streptomycin, and 100 U/mL penicillin G sodium, in a 37 °C incubator with 5% CO_2_. When the cells reached approximately 70% confluence, they were detached using 0.25% trypsin-EDTA and passage into new culture dishes with a diameter of 100 mm. For all senescence attenuation experiments, hMSCs at passage 6 were used. One day prior to experiments, the culture medium was replaced with α-MEM containing 1% FBS.

### 4.3. Cell Proliferation Assay

The proliferative effect of FN9-10ELP on cells was evaluated using the MTT assay [3-(4,5-dimethylthiazol-2-yl)-2,5-diphenyltetrazolium bromide; AMRESCO Inc., Solon, OH, USA]. This method was performed according to the protocol described in the cited study [17,18,45]. The MTT assay is a quantitative method that indirectly reflects cell viability and proliferation based on mitochondrial function. FN9-10ELP was coated onto 4-well plates at a concentration of 1 μg/cm^2^, based on previous study [17], demonstrating that this concentration provides optimal surface coverage to enhance cell adhesion and proliferation while avoiding surface oversaturation. Plates were incubated overnight at 4 °C prior to cell seeding. hMSCs were seeded onto both the FN9-10ELP-coated plates and non-coated plates (control) at a density of 3 × 10^4^ cells/well and incubated for 1–2 h at 37 °C in a 5% CO_2_ incubator to allow for cell attachment. Subsequently, cellular senescence was induced by treating the cells with 20 μM etoposide (Sigma, St. Louis, MO, USA), a concentration to efficiently induce premature senescence in hMSCs without causing excessive cytotoxicity, thereby preserving a viable cell population for downstream analyses [46]. Cells were cultured for 0, 1, and 2 days, where 0 day refers to the day of cell seeding on which the MTT assay was performed. At each time point, cells were washed with Dulbecco’s phosphate-buffered saline (DPBS), followed by the addition of 50 μL of 5 mg/mL MTT solution to each well. The plates were incubated for 1 h at 37 °C in a 5% CO_2_ incubator. After incubation, the MTT solution was removed, and 100 μL of dimethyl sulfoxide (DMSO) was added to each well to dissolve the formazan crystals for 30 min. Cell viability was quantified by measuring absorbance at 540 nm using a microplate reader (Biotech, Seoul, Republic of Korea). No reference wavelength was used, as FN9-10ELP coating did not visibly interfere with optical density measurements, and blank wells without cells were included to control for potential background absorbance.

### 4.4. RNA Extraction and cDNA Synthesis

Non-senescent hMSCs were cultured for 2 days on 100 mm culture dishes non-coated (control) and coated with FN9-10ELP at a concentration of 1 μg/cm^2^. Total RNA was extracted using the Easy-spinTM Total RNA Extraction Kit (iNtRON Biotechnology, Seoul, Republic of Korea) according to the manufacturer’s instructions. The purity and concentration of the extracted RNA were assessed by measuring absorbance at 260 nm and 280 nm using a NanoDrop 2000 spectrophotometer (Thermo Scientific, Waltham, MA, USA). Only RNA samples with an A260/A280 ratio between 1.9 and 2.1 were selected for subsequent analysis.

For reverse transcription, 2 μg of total RNA was used in a 20 μL reaction mixture. The reaction mixture contained 2 μL of 10× RT random primer (Invitrogen, Carlsbad, CA, USA), 0.6 μL of 25× dNTP mix (Invitrogen), 2 μL of enzyme buffer (Invitrogen), and 1 μL of M-MLV Reverse Transcriptase (Invitrogen). The reverse transcription reaction was performed at 37 °C for 2 h, following the manufacturer’s protocol (Applied Biosystems, Waltham, MA, USA), and the enzyme was subsequently inactivated at 85 °C for 5 min. The resulting complementary DNA (cDNA) was stored at −20 °C until further use.

### 4.5. Quantitative PCR Analysis

The mRNA expression levels of senescence-related genes (*IL-6*, *IL-8*, *PAI-1*) and the internal control gene *β-actin* were analyzed by quantitative polymerase chain reaction (qPCR). The primer sequences for each gene are listed in Table 1. All qRT-PCR reactions were performed using the StepOne^TM^ Real-Time PCR System (Applied Biosystems).

Each PCR reaction was carried out in a total volume of 15 μL, containing template cDNA, gene-specific primers, and 2× qPCR SYBR Green Master Mix (Cat. No. T22O4V2, Applied Biosystems).

The thermal cycling conditions were as follows: initial enzyme activation at 95 °C for 1 min, followed by 45 cycles of denaturation at 95 °C for 15 s, annealing at 55 °C for 15 s, and extension at 72 °C for 45 s. The threshold cycle (Ct) values obtained after amplification were used for the quantitative analysis of gene expression.

### 4.6. DAPI Staining and Nuclear Size Analysis

FN9-10ELP was applied to 12-well plates at a concentration of 1 μg/cm^2^ and coated overnight at 4 °C. hMSCs were seeded onto FN9-10ELP-coated and non-coated control wells at a density of 2 × 10^5^ cells per well and incubated at 37 °C in a 5% CO_2_ incubator for 1–2 h to allow cell attachment. Etoposide was then added at a final concentration of 20 μM to induce cellular senescence, and cells were cultured for 5 days.

Cultured cells were fixed with 3.7% formaldehyde for 10 min at 37 °C in a 5% CO_2_ incubator and then washed with DPBS. Permeabilization was performed using 0.5% Tri-ton X-100 for 15 min, followed by another wash with DPBS. The fixed and permeabilized cells were stained with DAPI (Sigma, St. Louis, MO, USA) solution for 15 min and subsequently washed three times with DPBS.

Fluorescent images were captured using an inverted fluorescence microscope (Olympus IX83, Olympus Corp., Tokyo, Japan), and nuclear size was analyzed using Fiji software (based on ImageJ; NIH, Bethesda, MD, USA, https://imagej.net/software/fiji/downloads, accessed on 12 May 2025).

### 4.7. Senescence-Associated β-Galactosidase (SA-β-Gal) Staining

SA-β-gal staining was performed using the Cell Senescence β-Galactosidase Staining Kit (MCE, HY-K1089, Monmouth Junction, NJ, USA) according to the manufacturer’s protocol to analyze SA-β-gal activity in cultured cells. FN9-10ELP was applied to 12-well plates at a concentration of 1 μg/cm^2^ and coated overnight at 4 °C.

Subsequently, hMSCs were seeded onto the FN9-10ELP-coated plates and non-coated control plates at a density of 5 × 10^4^ cells/well, and incubated at 37 °C in a 5% CO_2_ incubator for 1–2 h to allow cell attachment. Cellular senescence was induced by treating the cells with 20 μM etoposide, and the cells were cultured for 5 days.

SA-β-gal-positive cells were observed using a bright-field microscope at 20× magnification. The total number of cells was determined by DAPI staining, and the percentage of SA-β-gal-positive cells was calculated using the following formula: (Number of positive cells/Total number of cells) × 100.

### 4.8. Statistical Analysis

All experiments were performed with at least three independent biological replicates. Data are presented as mean ± standard deviation (SD). Statistical significance between two groups was analyzed using Welch’s *t*-test, and a *p*-value less than 0.05 was considered statistically significant. All statistical analyses were performed using R statistical software (R Project for Statistical Computing, version 4.4.3, Vienna, Austria, https://www.r-project.org, accessed on 8 April 2025). Statistical significance is indicated as follows: *p* < 0.05 (*), *p* < 0.01 (**), *p* < 0.001 (***), and *p* < 0.0001 (****).

## 5. Conclusions

This study demonstrates that FN9-10ELP, a recombinant ECM-mimetic fusion protein, effectively suppresses etoposide-induced cellular senescence in hMSCs. FN9-10ELP not only improves cell viability but also downregulates key senescence-associated genes (*IL-6*, *IL-8*, *PAI-1*), reduces nuclear enlargement, and significantly lowers SA-β-gal activity. These anti-senescent effects are consistent with our previous findings under normal, untreated culture conditions, where FN9-10ELP was shown to enhance cell adhesion, proliferation, and maintenance of the hMSC phenotype [17]. Together with its structural stability and biofunctionality, these findings suggest that FN9-10ELP holds considerable potential as a functional biomaterial for enhancing stem cell viability and delaying senescence. Future studies should further investigate its molecular mechanisms, including the involvement of modulation of the cellular microenvironment and integrin-mediated signaling pathways such as FAK/AKT/MAPK, and evaluate its efficacy in long-term cultures and in vivo regenerative models to comprehensively validate its translational potential.

## Figures and Tables

**Figure 1 ijms-26-09218-f001:**
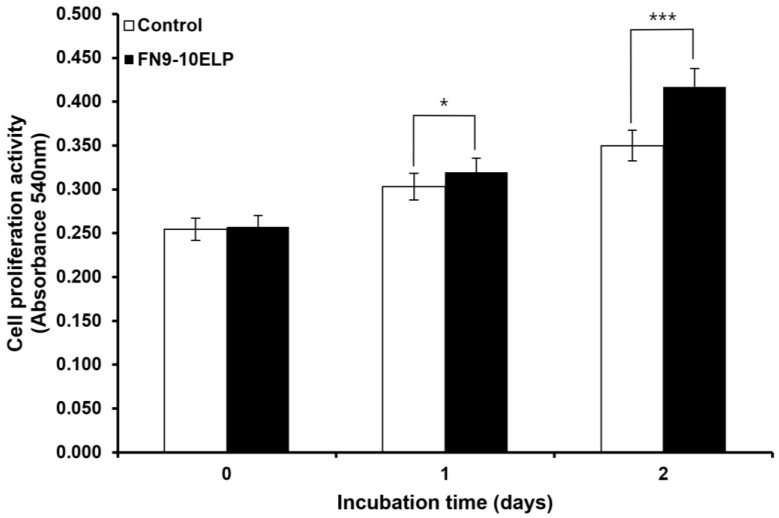
Analysis of hMSCs proliferative potential under etoposide (20 μM) treatment in the presence or absence of FN9-10ELP. hMSCs were cultured under two conditions: with FN9-10ELP coating (1 μg/cm^2^) and non-coating (control). Cellular senescence was induced by treating both groups with etoposide (20 μM) for 2 days. MTT assays were performed on days 0, 1, and 2, and absorbance measurements were used as an indicator of cell proliferation. The FN9-10ELP group showed significantly higher optical density, and thus higher cell number, than the control group from day 1, and the difference became more pronounced on day 2. Data are presented as mean ± SD (*n* = 4), and statistical significance was determined using an unpaired, two-tailed Welch’s *t*-test (* *p* < 0.05, *** *p* < 0.001).

**Figure 2 ijms-26-09218-f002:**
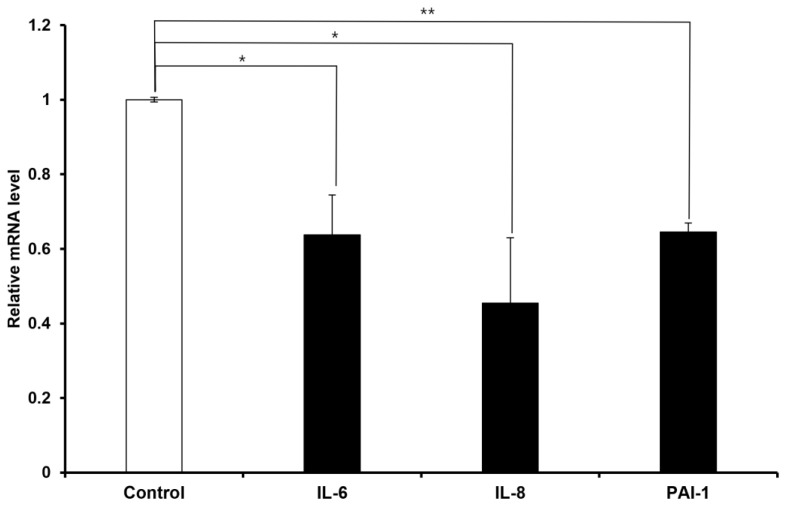
FN9-10ELP suppresses the expression of senescence-associated genes in etoposide-treated hMSCs. To induce senescence, hMSCs were seeded onto FN9-10ELP-coated or control dishes and treated with 20 μM etoposide to assess the preventive effect of FN9-10ELP on senescence. The mRNA levels of *IL-6*, *IL-8*, and *PAI-1* were analyzed by real-time PCR, and the expression levels were normalized to *β-actin* and presented as relative expression. All three genes were significantly downregulated in the FN9-10ELP group, indicating that FN9-10ELP attenuates the transcriptional activation of senescence-associated genes. Data are presented as mean ± SD (*n* = 3), and statistical significance was determined using an unpaired, two-tailed Welch’s *t*-test (* *p* < 0.05, ** *p* < 0.01). Open bars indicate control (uncoated dishes), and closed bars indicate FN9-10ELP-coated dishes.

**Figure 3 ijms-26-09218-f003:**
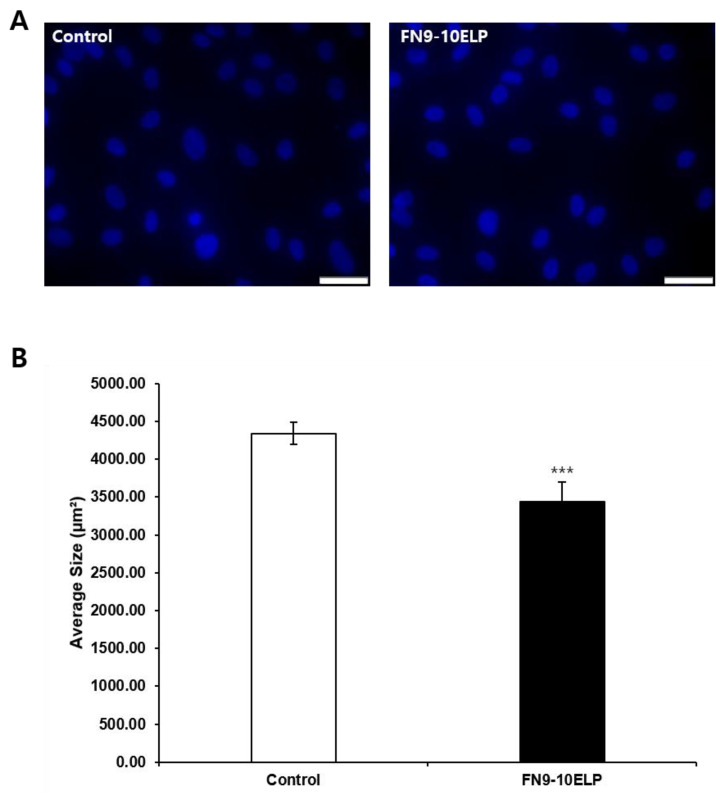
FN9-10ELP attenuates etoposide-induced nuclear enlargement in hMSCs. (**A**) Representative fluorescence images of DAPI-stained hMSCs treated with 20 μM etoposide in control and FN9-10ELP-coated conditions (scale bar: 50 μm). DAPI staining visualized nuclear morphology, including senescence-associated changes such as enlargement and altered chromatin organization. Although qualitative differences may appear subtle at this magnification, quantitative analysis revealed significant differences. (**B**) Quantification of nuclear area (μm^2^) based on at least 100 randomly selected nuclei per condition from three independent experiments. After 5 days of etoposide treatment, the control group exhibited a significantly larger nuclear area (4338.45 μm^2^) compared to the FN9-10ELP group (3438.61 μm^2^). Data are presented as mean ± SD (*n* = 5), and statistical significance was determined using Welch’s *t*-test (*** *p* < 0.001). Open bars indicate control (uncoated dishes), and closed bars indicate FN9-10ELP-coated dishes.

**Figure 4 ijms-26-09218-f004:**
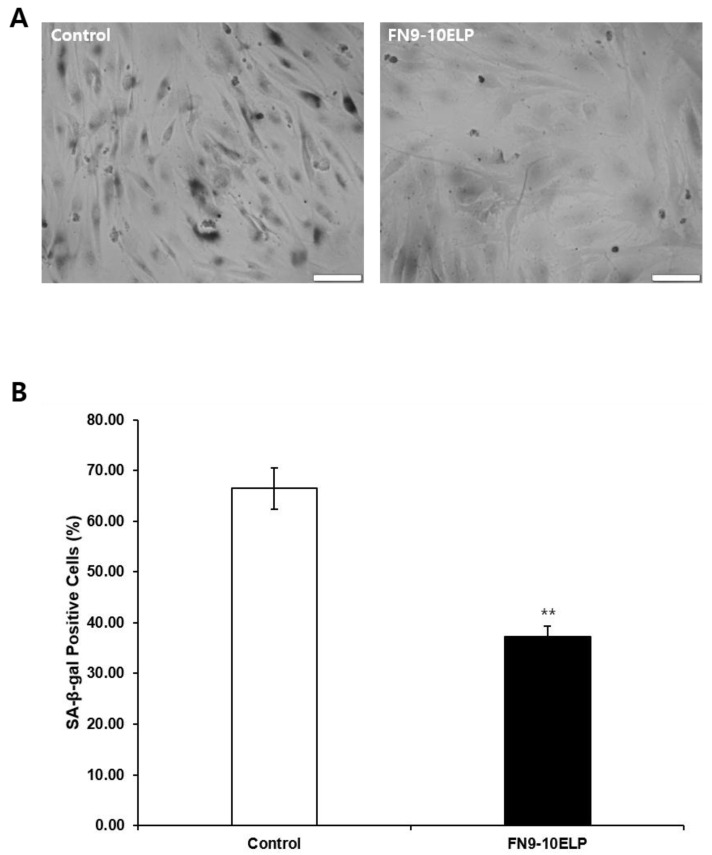
FN9-10ELP attenuates etoposide-induced cellular senescence in hMSCs. (**A**) Representative images of SA-β-gal-stained hMSCs treated with 20 μM etoposide for 5 days under control or FN9-10ELP-coated conditions. (scale bar: 100 μm). (**B**) Quantification of SA-β-gal-positive cells (blue cytoplasmic staining), expressed as a percentage of total cells. At least 200 cells per condition were analyzed using Fiji software (ImageJ; NIH, Bethesda, MD, USA), from three independent experiments. Data are presented as mean ± SD (*n* = 3), and statistical significance was determined using Welch’s *t*-test (** *p* < 0.01). Open bars indicate control (uncoated dishes), and closed bars indicate FN9-10ELP-coated dishes.

**Table 1 ijms-26-09218-t001:** Sequence of primer used in the real-time PCR.

Genes	Forward Primer	Reverse Primer
*β-actin*	TGGCACCCAGCACAATGAAGAT	TACTCCTGCTTGCTGATCCA
*IL-6*	CCCCTGACCCAACCACAAAT	GCCCAGTGGACAGGTTTCTG
*IL-8*	GTCTGCTAGCCAGGATCCAC	AGTGCTTCCACATGTCCTCA
*PAI-1*	CAGACCAAGAGCCTCTCCAC	GGTTCCATCACTTGGCCCAT

## Data Availability

The original data presented in the study are openly available in FigShare at DOI/10.6084/m9.figshare.30006322, accessed on 28 August 2025.

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
