# Peer review of "FN9-10ELP, an ECM-Mimetic Fusion Protein, Protects Human Mesenchymal Stem Cells from Etoposide-Induced Senescence"

_ijms, 2025, doi:10.3390/ijms26189218_

Round 1

Reviewer 1 Report

Comments and Suggestions for Authors

In this study the authors present the ECM-mimetic fusion protein FN9-10ELP as a promising anti-senescent agent in etoposide-induced senescent human turbinate-derived MSCs.

The study is the continuation of a previous work (reference 17), but I am afraid that it is mainly descriptive, providing no data on the mode of the anti-senescence action of the protein of interest. Most importantly, I have some major concerns on the experimental design and the conclusions drawn regarding senescence induction and senescence prevention.  

Major points

  1. Please check throughout the text parts referring to viability and senescence. In most cases, the term viability should be replaced by the more appropriate term proliferation. Given that no cell death trigger is used and given that senescence is not accompanied by cell death (on the contrary, senescent cells are apoptosis-resistant), differences in the color intensity of the formazan crystals corresponding to cell numbers in the MTT assay most probably reflect differences in cell proliferation rather than differences in cell viability. This is adequately described in the Materials and Methods section, but not in the rest of the text (please refer also to the uploaded file).
  2. Line 101: The authors should explain what day 0 exactly means. Were the cells pre-cultured onto FN9-10ELP at day 0? Or do the authors just refer to the starting material of the experiment at t=0? If the latter is correct, then referring to similar initial cell attachment is not physiologically relevant. This is better explained at the end of the manuscript, in the Materials and Methods section (If I am correct, time starts to count 2 h post-seeding of the cells onto the FN9-10ELP coated wells), but it should also be explained here.
  3. The authors should clarify if senescence induction preceded seeding of the cells onto FN9-10ELP-coated culture dishes. In line 134, the opposite from what is described above in the textis written. Were the cells first driven to senescence and then plated to FN9-10ELP-coated culture dishes or first plated to FN9-10ELP-coated culture dishes and then exposed to etoposide? I believe that this is the case but this should be clarified.
  4. If cells were first plated onto FN9-10ELP-coated culture dishes and then driven to senescence, then any observed action of FN9-10ELP would be preventive. Experiments should be repeated using cells that were first driven to senescence and then grown in the presence of FN9-10ELP to assess if an already established (pre-)senescent phenotype could be ameliorated by FN9-10ELP.
  5. Line 356: Assuming that FN9-10ELP coating was not dissolved by DMSO influencing optical density, was absorbance measured at a reference wavelength, as well, to normalize for putative differences in the absorbance of coated vs non-coated wells independent of the cells' presence?
  6. Lines 430-432: This is entirely speculative. There is no evidence or at least indication so as to draw this conclusion.
  7. My most important concern: It seems that control sample is an already senescent population, but this could not be confirmed in the absence of the young counterparts of the cells. In all experiments two samples are missing: untreated and FN9-10ELP-treated YOUNG cells as references. These are the required controls in order to confirm senescence induction, to be able to estimate the degree of anti-senescence action and to validate that the observed activity id indeed anti-senescence action and not a general anti-inflammatory effect (please also refer to the uploaded file).

Minor points

  1. The text contains some grammar and syntax errors that need to be corrected (for examples, please refer to the uploaded file).
  2. Throughout the text: real time and quantitative are redundant terms. RT in RT-qPCR stands for reverse transcription. Please check and correct accordingly.
  3. Lines 62-64: Was this done in the current study or in studies cited as 17 and 18? This should be clarified.
  4. Was incubation with etoposide was for 2 or 5 days? Please clarify (please also refer to the uploaded file).
  5. Line 178: This cannot be inferred given the absence of a young control sample.
  6. Line 341: Please replace by mitochondrial function.
  7. Lines 351-354 and line 370: Please check concentration and incubation times.
  8. Line 399: Please mention that this is an inverted fluorescence microscope.

Author Response

[Major points]

Reviewer Comment:

  1. Please check throughout the text parts referring to viability and senescence. In most cases, the term viability should be replaced by the more appropriate term proliferation. Given that no cell death trigger is used and given that senescence is not accompanied by cell death (on the contrary, senescent cells are apoptosis-resistant), differences in the color intensity of the formazan crystals corresponding to cell numbers in the MTT assay most probably reflect differences in cell proliferation rather than differences in cell viability. This is adequately described in the Materials and Methods section, but not in the rest of the text.

Response:

We thank the Reviewer for this important and constructive comment. We agree that, since no cell death trigger was applied and senescent cells are apoptosis-resistant, the differences observed in the MTT assay should be more accurately interpreted as changes in cell proliferation rather than viability. Accordingly, we have revised the terminology throughout the manuscript, including the Results section title and text (e.g., Section 2.1, Discussion), to consistently use the term “proliferation” instead of “viability,” except where cell viability is explicitly relevant. This correction improves the accuracy and clarity of our interpretation.

The revised sentence now reads:

2.1. FN9-10ELP Protects hMSCs from Etoposide-Induced Cellular Senescence by Promoting Proliferation

To evaluate whether the recombinant ECM-mimetic fusion protein FN9-10ELP can attenuate etoposide-induced senescence in hMSCs, cell proliferation was assessed using the MTT assay (Figure 1).

These results demonstrate that FN9-10ELP effectively enhances the proliferation of hMSCs in atime-dependent manner under etoposide-induced senescence conditions and indirectly suggest its role in attenuating cellular senescence.

Figure 1. Analysis of hMSCs proliferative potential under etoposide (20 μM) treatment in the presence or absence of FN9-10ELP. hMSCs were cultured under two conditions: with FN9-10ELP coating (1 μg/cm2) and non-coating (control). Cellular senescence was induced by treating both groups with etoposide (20 μM). MTT assays were performed on days 0, 1, and 2, and absorbance measurements were used as an indicator of cell proliferation. The FN9-10ELP group showed significantly higher optical density, and thus higher cell number, than the control group from day 1, and the difference became more pronounced on day 2.

Nevertheless, this study demonstrates that FN9-10ELP can effectively suppress hMSCs senescence and promote cell proliferation.

Reviewer Comment:

  1. Line 101: The authors should explain what day 0 exactly means. Were the cells pre-cultured onto FN9-10ELP at day 0? Or do the authors just refer to the starting material of the experiment at t=0? If the latter is correct, then referring to similar initial cell attachment is not physiologically relevant. This is better explained at the end of the manuscript, in the Materials and Methods section (If I am correct, time starts to count 2 h post-seeding of the cells onto the FN9-10ELP coated wells), but it should also be explained here.

Response:

We thank the Reviewer for this important comment. To clarify, day 0 refers to the time point immediately after seeding the hMSCs onto FN9-10ELP-coated or uncoated control wells, followed by etoposide treatment, and the MTT assay was performed 1 hour post-treatment. We have updated the Results section to include this explanation, and the statement regarding “similar initial cell attachment” has been revised to indicate comparable initial seeding conditions rather than implying physiologically established adhesion. This ensures accurate interpretation of the early time point data.

The revised sentence now reads:

On day 0, cells were seeded onto FN9-10ELP-coated or uncoated control group, followed by etoposide treatment, and the MTT assay was performed 1 hour post-treatment. At this early time point, there was no significant difference in absorbance between the FN9-10ELP group and the control group, indicating comparable initial seeding conditions rather than established cell adhesion.

Reviewer Comment:

  1. The authors should clarify if senescence induction preceded seeding of the cells onto FN9-10ELP-coated culture dishes. In line 134, the opposite from what is described above in the textis written. Were the cells first driven to senescence and then plated to FN9-10ELP-coated culture dishes or first plated to FN9-10ELP-coated culture dishes and then exposed to etoposide? I believe that this is the case but this should be clarified.

Response:

We appreciate the reviewer’s comment and clarification request. In our study, hMSCs were first seeded onto FN9-10ELP-coated or uncoated control dishes, and then treated with 20 μM etoposide to induce senescence. Therefore, the FN9-10ELP was present during etoposide exposure, allowing us to specifically evaluate its preventive effect against the induction of senescence. We have revised the text in the Results section (Figure 2 legend) to clearly describe this sequence of events.

The revised sentence now reads:

Figure 2. FN9-10ELP suppresses the expression of senescence-associated genes in etoposide-treated hMSCs. To induce senescence, hMSCs were seeded onto FN9-10ELP-coated or control dishes and treated with 20 μM etoposide to assess the preventive effect of FN9-10ELP on senescence.

Reviewer Comment:

  1. If cells were first plated onto FN9-10ELP-coated culture dishes and then driven to senescence, then any observed action of FN9-10ELP would be preventive. Experiments should be repeated using cells that were first driven to senescence and then grown in the presence of FN9-10ELP to assess if an already established (pre-)senescent phenotype could be ameliorated by FN9-10ELP.

Response:

We thank the reviewer for this important suggestion. In the current study, hMSCs were indeed seeded onto FN9-10ELP-coated dishes prior to etoposide treatment, allowing us to specifically assess the preventive effect of FN9-10ELP against senescence induction. We agree that experiments using pre-senescent cells would be valuable to evaluate whether FN9-10ELP can reverse an established senescent phenotype. This point has been added to the Discussion section as a limitation and a direction for future studies.

The revised sentence now reads:

However, this study has limitations. To assess the preventive effect of FN9-10ELP on etoposide-induced senescence, hMSCs were first seeded onto FN9-10ELP-coated or uncoated control dishes, followed by treatment with 20 μM etoposide. This approach allowed us to specifically evaluate the preventive action of FN9-10ELP against senescence induction. We acknowledge, however, that this experimental design does not allow us to assess whether FN9-10ELP can reverse an already established senescent phenotype. Future studies using pre-senescent hMSCs, in which cells are first driven to senescence and then cultured in the presence of FN9-10ELP, would be valuable to determine whether FN9-10ELP can ameliorate established senescence. Furthermore, additional research is needed to clarify the molecular mechanisms of FN9-10ELP, particularly its effects on integrin and downstream signaling pathways. In addition, the long-term impact of FN9-10ELP on stem cells stemness and differentiation potential remains to be investigated.

Reviewer Comment:

  1. Line 356: Assuming that FN9-10ELP coating was not dissolved by DMSO influencing optical density, was absorbance measured at a reference wavelength, as well, to normalize for putative differences in the absorbance of coated vs non-coated wells independent of the cells' presence?

Response:

We appreciate the reviewer’s comment. In our experiments, absorbance was measured at 540 nm without using a reference wavelength. FN9-10ELP coating did not visually or experimentally interfere with optical density measurements, and blank wells without cells were included in each condition to account for any background absorbance. This approach ensured that the measured absorbance accurately reflected formazan production corresponding to cell proliferation.

The revised sentence now reads:

Cell viability was quantified by measuring absorbance at 540 nm using a microplate reader. No reference wavelength was used, as FN9-10ELP coating did not visibly interfere with optical density measurements, and blank wells without cells were included to control for potential background absorbance.

Reviewer Comment:

  1. Lines 430-432: This is entirely speculative. There is no evidence or at least indication so as to draw this conclusion.

Response:

We thank the reviewer for this comment. To address the concern regarding speculative statements, we have revised the Conclusions section to clearly indicate that, while FN9-10ELP shows considerable potential for enhancing stem cell viability and delaying senescence, further studies are needed to experimentally confirm the molecular mechanisms, including the potential involvement of cellular microenvironment modulation and integrin-mediated signaling pathways (FAK/AKT/MAPK).

The revised sentence now reads:

Together with its structural stability and biofunctionality, these findings suggest that FN9-10ELP holds considerable potential as a functional biomaterial for enhancing stem cell viability and delaying senescence. Future studies should further investigate the molecular mechanisms, including the involvement of modulation of the cellular microenvironment and integrin-mediated signaling pathways such as FAK/AKT/MAPK, and also evaluate its efficacy in long-term cultures and in vivo regenerative models to comprehensively validate its translational potential.

Reviewer Comment:

  1. My most important concern: It seems that control sample is an already senescent population, but this could not be confirmed in the absence of the young counterparts of the cells. In all experiments two samples are missing: untreated and FN9-10ELP-treated YOUNG cells as references. These are the required controls in order to confirm senescence induction, to be able to estimate the degree of anti-senescence action and to validate that the observed activity id indeed anti-senescence action and not a general anti-inflammatory effect (please also refer to the uploaded file).

Response:

We thank the reviewer for this important comment. In our study, etoposide-untreated young hMSCs have been extensively characterized in our previous work [17], confirming baseline levels of senescence-associated markers. In the present experiments, our specific aim was to evaluate whether FN9-10ELP can prevent or attenuate etoposide-induced senescence. To this end, cells of the same passage were seeded onto either FN9-10ELP-coated or uncoated dishes and subsequently treated with etoposide to induce senescence. This experimental design allowed us to directly assess the preventive and anti-senescent effects of FN9-10ELP on induced senescence, using uncoated etoposide-treated cells as the internal control. We agree that inclusion of untreated young cells as an additional reference could further strengthen the comparative analysis; however, as the baseline characteristics of young hMSCs have already been well established in previous studies, the primary objective here was to investigate the modulation of experimentally induced senescence by FN9-10ELP. We have now added this clarification to the Discussion section.

(17. Lee, S., J. E. Kim, H. J. Seo, and J. H. Jang. "Design of Fibronectin Type Iii Domains Fused to an Elastin-Like Polypeptide for the Osteogenic Differentiation of Human Mesenchymal Stem Cells." Acta Biochim Biophys Sin (Shanghai) 51, no. 8 (2019): 856–63.)

The revised sentence now reads:

However, this study has limitations. To assess the preventive effect of FN9-10ELP on etoposide-induced senescence, hMSCs were first seeded onto FN9-10ELP-coated or un-coated control dishes, followed by treatment with 20 μM etoposide. This approach allowed us to specifically evaluate the preventive action of FN9-10ELP against senescence induction. In this study, all experimental groups were subjected to etoposide treatment to induce senescence, and the effects of FN9-10ELP were evaluated under these conditions. This design enabled us to specifically assess the preventive effect of FN9-10ELP against DNA damage-induced senescence. Untreated young hMSCs were not included as additional controls, since their baseline characteristics and responses under normal culture conditions have already been established in our previous studies [17]. Nonetheless, the absence of young counterparts in the present work is acknowledged as a limitation, and future studies should incorporate untreated and FN9-10ELP-treated young hMSCs to more comprehensively distinguish anti-senescence effects from general anti-inflammatory activity. We acknowledge, however, that this experimental design does not allow us to assess whether FN9-10ELP can reverse an already established senescent phenotype. Future studies using pre-senescent hMSCs, in which cells are first driven to senescence and then cultured in the presence of FN9-10ELP, would be valuable to determine whether FN9-10ELP can ameliorate established senescence. Furthermore, additional research is needed to clarify the molecular mechanisms of FN9-10ELP, particularly its effects on integrin and downstream signaling pathways. In addition, the long-term impact of FN9-10ELP on stem cells stemness and differentiation potential remains to be investigated.

[Minor points]

Reviewer Comment:

  1. The text contains some grammar and syntax errors that need to be corrected (for examples, please refer to the uploaded file).

Response:

We thank the reviewer for pointing this out. We have thoroughly checked the manuscript and corrected all grammar, syntax, and typographical errors. Sentences have been revised to improve clarity and readability, ensuring that the scientific content is accurately and effectively communicated.

The revised sentence now reads:

Abstract

Premature senescence was induced by treatment with 20 µM etoposide, and the protective effects of FN9-10ELP were evaluated in terms of cell viability (using the MTT assay), senescence-associated gene expression (by RT-qPCR analysis), nuclear morphology (after staining with DAPI), and SA-β-galactosidase activity.

Figure 1. ~ with FN9-10ELP coating (1 μg/cm2) and non-coating (control).

Reviewer Comment:

  1. Throughout the text: real time and quantitative are redundant terms. RT in RT-qPCR stands for reverse transcription. Please check and correct accordingly.

Response:

We thank the reviewer for pointing this out. The text has been revised to correctly refer to RT-qPCR, acknowledging that RT stands for reverse transcription, and removing the redundant use of “real-time” and “quantitative” throughout the manuscript.

The revised sentence now reads:

Abstract

senescence-associated gene expression (by RT-qPCR analysis),

Discussion

RT-qPCR analysis revealed that the expression of senescence-associated genes—IL-6, IL-8, and PAI-1—

4.5. Quantitative PCR Analysis

The mRNA expression levels of senescence-related genes (IL-6, IL-8, PAI-1) and the internal control gene β-actin were analyzed by quantitative polymerase chain reaction (qPCR).

Reviewer Comment:

  1. Lines 62-64: Was this done in the current study or in studies cited as 17 and 18? This should be clarified.

Response:

We thank the reviewer for the comment. The text has been revised to clarify that the expression and purification of the FN9-10ELP fusion protein, as described, were performed in the previous study cited as reference [17]. Reference [18] has been removed to avoid confusion.

The revised sentence now reads:

The FN9-10 domains were derived from human fibronectin, and the ELP sequence was de novo designed based on consensus elastin motifs, was expressed in Escherichia coli via re-combinant DNA technology, and was purified using inverse transition cycling [17].

Reviewer Comment:

  1. Was incubation with etoposide was for 2 or 5 days? Please clarify (please also refer to the uploaded file).

Response:

We thank the reviewer for this comment. Cellular senescence was induced by treating both groups with 20 μM etoposide for 2 days. This clarification has been added.

The revised sentence now reads:

Figure 1. Cellular senescence was induced by treating both groups with etoposide (20 μM) for 2 days.

Reviewer Comment:

  1. Line 178: This cannot be inferred given the absence of a young control sample.

Response:

We thank the reviewer for this comment. In our previous studies, the baseline characteristics of young hMSCs were well established. The current experiments specifically focused on evaluating whether FN9-10ELP could prevent or attenuate etoposide-induced senescence when compared to non-coated FN9-10ELP control cells. To this end, cells of the same passage were seeded onto either FN9-10ELP-coated or uncoated culture dishes and subsequently treated with etoposide. This design allowed us to use etoposide-treated cells without FN9-10ELP as an internal control and directly assess the preventive and anti-senescent effects of FN9-10ELP on induced senescence.

Reviewer Comment:

  1. Line 341: Please replace by mitochondrial function.

Response:

We thank the reviewer for the suggestion. The text has been revised to clarify that the MTT assay reflects mitochondrial function, rather than general cell viability or proliferation, based on cellular metabolic activity.

The revised sentence now reads:

4.3. Cell proliferation assay

The MTT assay is a quantitative method that indirectly reflects cell viability and proliferation based on mitochondrial function.

Reviewer Comment:

  1. Lines 351-354 and line 370: Please check concentration and incubation times.

Response:

We thank the reviewer for the comment.

Lines 351-354

- 0 day: This refers to the same day of cell seeding, immediately after the cells were attached to the FN9-10ELP-coated wells, consistent with our experimental protocol.

- MTT concentration (5 mg/mL) and incubation time (1 hour): Both the concentration and incubation time were adopted from previously established protocols. These conditions allow reliable measurement of formazan production without affecting cell viability. The relevant reference has been cited accordingly.

Line 370

- 2hours: The reverse transcription step was conducted at 37 °C for 2 hours in accordance with the Applied Biosystems protocol. The text has been revised to clarify this.

We have added this clarification in the Methods section to ensure transparency regarding the timing, concentration, and incubation conditions.

The revised sentence now reads:

4.3. Cell proliferation assay

The proliferative effect of FN9-10ELP on cells was evaluated using the MTT assay [3-(4,5-dimethylthiazol-2-yl)-2,5-diphenyltetrazolium bromide; AMRESCO Inc., Solon, USA]. This method was performed according to the protocol described in the cited study [17, 18, 44].

Cells were cultured for 0, 1, and 2 days, where 0 day refers to the day of cell seeding on which the MTT assay was performed.

4.4. RNA Extraction and cDNA Synthesis

The reverse transcription reaction was performed at 37 ℃ for 2 hours, following the manufacturer’s protocol (Applied Biosystems, USA), and the enzyme was subsequently inactivated at 85 ℃ for 5 minutes.

Reviewer Comment:

  1. Line 399: Please mention that this is an inverted fluorescence microscope.

Response:

We thank the reviewer for this comment. We have clarified in the Methods section that an inverted fluorescence microscope was used to capture the fluorescent images.

The revised sentence now reads:

4.6. DAPI Staining and Nuclear Size Analysis

Fluorescent images were captured using an inverted fluorescence microscope (Olympus IX83, Olympus Corp., Tokyo, Japan),

[Other points]

Reviewer Comment:

Line 150: What was the area of untreated young cells?

Response:

We thank the reviewer for this insightful comment. In this study, the nuclear area of untreated young hMSCs was not measured. Our experimental design specifically aimed to assess whether FN9-10ELP could prevent or attenuate nuclear enlargement and senescence induced by etoposide when compared between coated and uncoated conditions. While our previous work has examined FN9-10ELP effects on untreated hMSCs, nuclear morphology such as nuclear area was not evaluated. We acknowledge that including untreated young cells as a reference would provide additional context, and we will consider this comparison in future studies.

Reviewer Comment:

Figure 3-A: We were unable to view the full original text, as only a portion of the sentence was available: “Control nuclei seem quite similar in terms of surface with those of FN9-10-treated cells, with only 2 or 3 …”

Response:

We thank the reviewer for this valuable observation. We agree that, in some representative images, the nuclear morphology of control cells may appear visually similar to that of FN9-10ELP-treated cells, with only subtle differences observable. However, quantitative analysis of nuclear area across multiple fields and replicates demonstrated a statistically significant reduction in nuclear enlargement in the FN9-10ELP group compared with the control group (4338.45 μm² vs. 3438.61 μm², p < 0.001). This indicates that, although the visual impression from selected images may suggest similarity, systematic measurement confirmed that FN9-10ELP effectively attenuated etoposide-induced nuclear enlargement.

Reviewer Comment:

Line 164, 172: Why etoposide treatment was longer in these experiments than the 2-days incubation described in the above-mentioned experiments?

Response:

We thank the reviewer for this comment. While MTT and RT-qPCR assays were conducted after 2 days of etoposide treatment, DAPI staining and SA-β-gal assays were performed at 5 days, as this time point yielded more meaningful and detectable changes in nuclear morphology and senescence-associated β-galactosidase activity. This approach allowed us to capture the most relevant anti-senescence effects of FN9-10ELP in these assays.

Reviewer Comment:

Line 358: ‘hMSCs’ were cultured for 2 days on 100mm culture…

Response:

We thank the reviewer for this comment. In our study, early-passage hMSCs (passage 6) were used. Cells were seeded onto FN9-10ELP-coated or uncoated control dishes, and after confirming cell attachment, etoposide treatment was applied to induce senescence. Therefore, the hMSCs were not already senescent prior to the experiments, and the design specifically allowed us to evaluate the preventive and anti-senescent effects of FN9-10ELP.

The revised sentence now reads:

Non-senescent hMSCs were cultured for 2 days on 100mm culture dishes non-coated (control) and coated with FN9-10ELP at a concentration of 1 μg/cm2.

Reviewer 2 Report

Comments and Suggestions for Authors

Senescence plays a significant role in the mechanisms of many serious pathologies. The study of anti-senescence can be useful for the design of new therapies. Nevertheless, there are several points that can be addressed to improve the impact and quality of the manuscript.

The mechanisms of senescence and the effects of anti-senescence should be better explained and discussed. What is the relationship between Akt signalling and the expression and activity of SA-β-galactosidase?

Chemotherapeutics such as etoposide can cause increased activity of NF-κB, as well as higher levels of IL-6 and IL-8, implying their possible roles in the process. Additionally, NF-κB can serve as a negative regulator of PTEN. Please consider this in our model.

Due to the role of oxidative stress, the intracellular levels of ROS should be determined.

Author Response

Reviewer Comment:

The mechanisms of senescence and the effects of anti-senescence should be better explained and discussed. What is the relationship between Akt signalling and the expression and activity of SA-β-galactosidase?

Response:

We thank the reviewer for the comment. We have added a discussion of the PI3K/AKT signaling pathway and its potential relationship with SA-β-gal activity in the revised manuscript to better explain the mechanisms of senescence and the anti-senescent effects of FN9-10ELP.

The revised sentence now reads:

Consistent with this, SA-β-gal staining revealed that the proportion of SA-β-gal-positive cells was significantly lower in the FN9-10ELP-coated group com-pared to the control (Figure 4). SA-β-galactosidase is a well-established biomarker of cellular senescence [33], and the reduced SA-β-gal activity observed here supports the anti-senescent effect of FN9-10ELP. Similarly, Tragoonlugkana et al. (2024) reported that adipose-derived stem cells (ADSCs) cultured on fibronectin (FN) or vitronectin (VN)-coated surfaces exhibited significantly fewer SA-β-gal-positive cells compared to non-coated controls [34]. Although different cell types were used, these findings are in agreement with our results and support the notion that ECM protein coatings can attenuate stem cells senescence. Hence, FN9-10ELP, an ECM-mimetic protein, demon-strates potential as a functional biomaterial for senescence attenuation. In addition, the PI3K/AKT signaling pathway is known to regulate lysosomal biogenesis and autophagy, key processes underlying SA-β-gal activity [38]. AKT activation regulates mTORC1 activity, which in turn prevents excessive lysosomal enlargement and maintains autophagic flux [39]. As mTORC1 functions as a central regulator of lysosomal capacity and autophagy, the activation of AKT may contribute to the normalization of lysosomal homeostasis and the preservation of autophagic balance in senescent cells [39]. Therefore, the observed reduction in SA-β-gal-positive cells following FN9-10ELP treatment may be partially attributable to integrin-mediated activation of AKT, which could potentially contribute to the restoration of lysosomal homeostasis and the mitigation of senescence-associated cellular alterations. These findings indirectly suggest a mechanistic connection between ECM-mediated signaling, AKT activation, and SA-β-gal activity in senescent cells.

Reviewer Comment:

Chemotherapeutics such as etoposide can cause increased activity of NF-κB, as well as higher levels of IL-6 and IL-8, implying their possible roles in the process. Additionally, NF-κB can serve as a negative regulator of PTEN. Please consider this in our model.

Response:

We thank the reviewer for the insightful comment. In response, we have considered the potential roles of NF-κB and its downstream cytokines, IL-6 and IL-8, in our model. Specifically, we acknowledge that chemotherapeutics such as etoposide may activate NF-κB signaling, which could contribute to cellular senescence, and that NF-κB can act as a negative regulator of PTEN. These points have been incorporated into the revised Discussion to provide a more comprehensive view of the molecular pathways potentially involved in senescence and the anti-senescent effects observed in our study.

The revised sentence now reads:

RT-qPCR analysis revealed that the expression of senescence-associated genes—IL-6, IL-8, and PAI-1—was significantly reduced in FN9-10ELP-coated hMSCs compared to non-coated controls (Figure 2). PAI-1 is a key factor known to amplify cellular senescence by enhancing the production of SASP components such as IL-6 and IL-8 [28], while IL-6 and IL-8 are representative SASP markers that are induced by DNA damage and contribute to proinflammatory microenvironments [29, 30]. Chemotherapeutic agents, such as etoposide, are known to activate NF-κB, which in turn increases the levels of IL-6 and IL-8 [31, 32]. Additionally, NF-κB can act as a negative regulator of PTEN, modulating downstream AKT signaling [33]. FN9-10ELP may suppress etoposide-induced NF-κB activation and SASP expression, thereby contributing to the reduction of senescence markers observed in hMSCs. Thus, this downregulation suggests that FN9-10ELP can alleviate cellular senescence by inhibiting inflammatory signaling pathways.

Reviewer Comment:

Due to the role of oxidative stress, the intracellular levels of ROS should be determined.

Response:

We thank the reviewer for this valuable suggestion. In line with the reviewer’s comment, we have revised the Discussion to acknowledge the role of ROS in etoposide-induced senescence. Specifically, we note that etoposide has been reported to induce DNA damage, mitochondrial biogenesis, and cytotoxicity through ROS generation [40]. Although intracellular ROS levels were not directly measured in our study, prior studies also show that ECM proteins containing fibronectin domains can attenuate ROS accumulation in MSCs under stress [41]. We have incorporated this information into the revised manuscript (Discussion) and cited the relevant references 40 and 41.

The revised sentence now reads:

Although we did not directly measure intracellular ROS levels in this study, previous research has shown that etoposide induces DNA damage, mitochondrial biogenesis, and cytotoxicity through excessive ROS generation [40]. Moreover, ECM proteins containing fibronectin domains similar to FN9-10 have been reported to modulate oxidative stress and reduce ROS accumulation in MSCs under stress conditions [41]. Taken together, these findings suggest that FN9-10ELP may help maintain redox homeostasis and protect hMSCs from oxidative stress-induced senescence, consistent with the anti-senescent effects observed in our study.

Round 2

Reviewer 1 Report

Comments and Suggestions for Authors

This is the revised version of a previously submitted manuscript.

The authors have addressed the majority of my concerns raised during the previous round of the reviewing process.

My main concern (about the absence of a treated and untreated young control) remains, but at least is now discussed by the authors as a limitation of the study.

Reviewer 2 Report

Comments and Suggestions for Authors

I have no objections